# Microbial Interkingdom Biofilms and the Quest for Novel Therapeutic Strategies

**DOI:** 10.3390/microorganisms9020412

**Published:** 2021-02-17

**Authors:** Katrien Van Dyck, Rita M. Pinto, Durgasruthi Pully, Patrick Van Dijck

**Affiliations:** 1Laboratory of Molecular Cell Biology, Institute of Botany and Microbiology, Department of Biology, KU Leuven, 3001 Leuven, Belgium; katrien.vandyck@kuleuven.be (K.V.D.); anaritaoliveiramacedo.pinto@kuleuven.be (R.M.P.); durgasruthi.pully@student.kuleuven.be (D.P.); 2VIB—KU Leuven Center for Microbiology, 3001 Leuven, Belgium; 3LAQV, REQUIMTE, Departamento de Ciências Químicas, Faculdade de Farmácia, Universidade Do Porto, 4050-313 Porto, Portugal

**Keywords:** polymicrobial, biofilm, *Candida*, biofilm matrix, quorum sensing, antimicrobial peptides, essential oils, nanoparticles, probiotics

## Abstract

Fungal and bacterial species interact with each other within polymicrobial biofilm communities in various niches of the human body. Interactions between these species can greatly affect human health and disease. Diseases caused by polymicrobial biofilms pose a major challenge in clinical settings because of their enhanced virulence and increased drug tolerance. Therefore, different approaches are being explored to treat fungal–bacterial biofilm infections. This review focuses on the main mechanisms involved in polymicrobial drug tolerance and the implications of the polymicrobial nature for the therapeutic treatment by highlighting clinically relevant fungal–bacterial interactions. Furthermore, innovative treatment strategies which specifically target polymicrobial biofilms are discussed.

## 1. Introduction

Within the human body, microorganisms mostly exist in complex communities, including bacteria, fungi, and viruses [1]. In various niches of the host, interactions between fungi and bacteria frequently occur during infections [2]. The most diverse polymicrobial niches are the oral cavity and gastrointestinal tract; however, polymicrobial diseases may be located throughout the human body [1]. Both mucosal tissues and abiotic surfaces, such as catheters, dentures, and implants, are ideal surfaces for polymicrobial biofilm formation. Although it is widely accepted that the human body contains tremendous microbial diversity, most research was focused on bacteria while the fungal species, referred to as the mycobiome, were neglected [2]. *Candida* species, of which *Candida albicans* is the most prevalent, are commensal fungal species often involved in biofilm-related infections [3]. *C. albicans* is an opportunistic pathogen able to cause a variety of infections mainly in immunocompromised patients [4]. Other *Candida* species, including *Candida glabrata*, *Candida tropicalis*, *Candida parapsilosis, Candida krusei*, and *Candida auris*, are emerging as important causes of fungal infections as well [5]. These *Candida* species exhibit varying degrees of intrinsic resistance to the commonly used antifungal agents. The shift toward intrinsically resistant species, such as *C. krusei* and *C. auris*, in the epidemiology of *Candida* infections poses an additional medical concern [6,7]. *Candida* biofilm-associated infections are also a major challenge in clinical settings due to their intrinsic resistance to antifungals and the host immune response [8]. However, interactions of *Candida* species with bacteria in polymicrobial biofilms are an even bigger hurdle as they greatly impact the efficiency of the treatment strategy and the disease outcome [9]. *C. albicans* co-exists with commensal bacteria in several niches of the host and some of these fungal–bacterial interactions have a clear synergistic interaction, enhancing the pathogenicity of one or both species.

*Staphylococcus aureus* is a Gram-positive commensal bacterium of healthy humans, but is also an opportunistic pathogen able to cause disease [10]. Among the wide variety of diseases ranging from superficial to life-threatening, methicillin-resistant *S. aureus* (MRSA) is an important cause of nosocomial infections [11,12]. *C. albicans* and *S. aureus* are often co-isolated from a variety of biofilm-associated diseases, including periodontitis, cystic fibrosis, and denture stomatitis, and they display a specific increased pathogenicity and enhanced drug tolerance [13,14,15]. A similar synergistic interaction is observed for *C. albicans* with *Staphylococcus epidermidis*, a Gram-positive commensal bacterium often involved in implant-associated infections [16]. Another bacterium often isolated together with *C. albicans,* especially from cystic fibrosis patients, is a Gram-negative bacterium *Pseudomonas aeruginosa* [17]. In contrast to the obviously synergistic interaction with *S. aureus*, the nature of the interaction with *P. aeruginosa* is not completely clear, and highly depends on the host and model system used to establish co-infection [18]. Streptococcal bacteria including *Streptococcus oralis*, *Streptococcus mutans*, *Streptococcus gordonii*, and *Streptococcus sanguinis* are the primary colonizers of the oral cavity. Interactions between oral bacteria and *C. albicans* tend to be mostly synergistic, stimulating the colonization and pathogenic potential of one or more microorganisms [19]. Several diseases of the oral cavity, including dental caries, endodontic infections, periodontal diseases, and denture-related infections, are associated with the formation of polymicrobial biofilms [20]. In addition, *C. albicans* provides a hypoxic microenvironment that supports the growth of anaerobic bacteria as well [21].

In clinical settings, polymicrobial diseases are increasingly recognized and are often accompanied with altered infection outcomes and therapeutic problems. The most obvious challenge in the treatment of interkingdom biofilms is the evolutionary distance between the disease-causing microorganisms, since the majority of antimicrobials target only one causative agent [22]. Standard treatment strategies for interkingdom polymicrobial infections, therefore, involve a combination of an antibacterial and antifungal drug. However, this approach has a poor efficacy, resulting in high chances of treatment failure [23,24]. The main reason is the specific increased drug tolerance and enhanced pathogenicity when the nature of the biofilm is polymicrobial, caused by a variety of factors including the biofilm matrix and quorum sensing [9]. Therefore, novel treatment strategies in which a sole agent can be used to prevent or eradicate polymicrobial biofilms are urgently needed. In this regard, alternative treatment strategies including antimicrobial peptides, plant-derived components, quorum quenchers, probiotics, and the use of nanoparticles are being explored for the treatment of fungal–bacterial biofilm infections.

In this review, the main mechanisms that underlie this specific polymicrobial pathogenicity and drug tolerance and the implications of the polymicrobial nature for the therapeutic treatment and infection outcome are described, using clinically relevant fungal–bacterial interactions. Furthermore, drawbacks of existing therapeutic strategies are raised, and novel treatment strategies are highlighted.

## 2. The Challenges of Targeting Interkingdom Biofilms

### 2.1. Enhanced Pathogenicity and Altered Infection Outcome

The synergistic interaction between fungal and bacterial species often leads to an altered infection outcome related with higher morbidity and mortality rates. For example, *C. albicans* and *S. aureus* display a synergistic interaction leading to enhanced drug resistance and increased mortality in different mouse models [23,25,26]. The most common fungal biofilm-like infection of the oral cavity is oropharyngeal candidiasis (OPC), predominantly caused by *C. albicans* [27]. Since *S. aureus* is also commonly isolated from the oral cavity, a mouse model of oral *C. albicans*–*S. aureus* co-infection was developed [28,29,30]. Using this model, it was found that co-infected mice had both a higher oral colonization of *S. aureus* and were susceptible to a systemic *S. aureus* infection, whilst this was not the case for mono-infected mice [29,30]. Afterwards, research was focused on figuring out the mechanism of *S. aureus* dissemination when co-infected with *C. albicans*. Thereby, two important requirements for bacterial dissemination were identified: the physical interaction between both species via *C. albicans* agglutinin-like sequence (Als) 1 and 3 and the host immune response [31,32,33]. This increased mortality and morbidity was also observed in other animal models. In a peritoneal infection mouse model, an infection with both *C. albicans* and *S. aureus* was lethal whilst the corresponding mono-microbial infections were not [26,34]. A similar synergistic interaction is observed for *C. albicans* with *S. epidermidis*, the most commonly co-isolated bacteria from *C. albicans* bloodstream infections [16]. Using a polymicrobial biofilm model in *Caenorhabditis elegans*, increased mortality was observed for a *C. albicans*–*S. epidermidis* co-infection compared to mono-microbial infections [35].

Regarding *P. aeruginosa*, the nature of the interaction with *C. albicans* is not completely clear. Several studies suggested that *P. aeruginosa* inhibits growth of *C. albicans* in vitro, thereby defining the interaction as antagonistic [18]. The physical interaction between the two species, mediated by type IV pili, appeared to be required for killing of *C. albicans* hyphae [36,37]. Similar effects were observed in co-infection mouse models of lung infections where prior colonization of *C. albicans* resulted in increased *P. aeruginosa* clearance [38,39]. However, in a zebrafish infection model, increased *C. albicans* pathogenicity was observed in a co-infection with *P. aeruginosa,* and also in a mouse burn wound model, higher mortality rates were observed in co-infected mice [40,41]. In addition, clinical observations also indicate a potential synergistic interaction as *Candida* colonization in the respiratory tract of critically ill patients receiving mechanical ventilation increased the risk of *P. aeruginosa* ventilator-associated pneumonia [42].

The interaction of *C. albicans* with oral streptococcal bacteria tends to be mostly synergistic. Biofilm formation of oral streptococcal bacteria including *S. gordonii*, *S. sanguinis*, and *S. oralis*, was promoted by *C. albicans* on the surface of an oral mucosa analogue [43]. The physical interaction of *C. albicans* with *S. gordonii* was shown to be mediated by streptococcal surface proteins SspA and SspB and the *Candida* adhesin Als3 [44,45,46]. The oral opportunistic pathogen *S. oralis* was found to colonize better in the presence of *C. albicans* in an oral infection mouse model [43,47]. Clearly, numerous animal studies confirm the increased persistence, severity, and mortality of fungal–bacterial co-infections. Additionally, in clinical settings, diseases including infections of the oral cavity, cystic fibrosis, and diabetic foot wound infections are recognized as polymicrobial infections and the composition of the microbial population can predict the disease severity and outcome of the infection [1].

### 2.2. The Biofilm Matrix: A Protective Shield

Biofilms are embedded in a matrix consisting of extracellular polysaccharides (EPS), proteins, and extracellular DNA (eDNA). Interestingly, more matrix material is produced in polymicrobial biofilms [23]. In *C. albicans*–*S. aureus* or *S. epidermidis* polymicrobial biofilms, the enhanced tolerance towards antibiotics originates from the extracellular matrix of the biofilm which limits drug penetration [10]. One of the main polysaccharides of the *C. albicans* biofilm matrix, β-1,3-glucan, was shown to be the key constituent for *S. aureus*-enhanced drug tolerance to vancomycin [48]. Likewise, extracellular polymers produced by *S. epidermidis* protected *C. albicans* against fluconazole by inhibiting penetration of the drug in a polymicrobial biofilm [49]. In addition, increased eDNA was observed in *C. albicans*–*S. epidermidis* or *S. aureus* mixed biofilms, and degradation of this eDNA significantly increased the sensitivity of *C. albicans* to miconazole [50,51].

*C. albicans* extracellular matrix components also play an important role in enhancing drug resistance of *P. aeruginosa* [52]. It was recently shown that *C. albicans* α-mannan and β-glucan polysaccharides increased the tolerance of *P. aeruginosa* to meropenem, directly impacting the treatment of co-infected cystic fibrosis patients [52]. In *P. aeruginosa–C. albicans* mixed biofilms, *C. albicans* enhances *P. aeruginosa* EPS production by increasing alginate-producing genes, *AlgU* and *mucA*. Thus, the synergy between these two microorganisms leads to thicker biofilms with an easier bacterial dissemination [53]. However, this effect seems to be attenuated in the presence of *N*-acetyl-l-cysteine both in vitro and in an in vivo catheter-subcutaneous implantation model. This thiol-containing cysteine derivative is commonly known to disrupt disulfide bonds and cysteine utilization, exhibiting antimicrobial activity [54]. Hence, *N*-acetyl-l-cysteine may be a promising agent to prevent biofilm formation and to attenuate catheter-related sepsis due to its ability to inhibit matrix production on *P. aeruginosa*–*C. albicans* biofilms [53].

In *C. albicans*–*S. mutans* polymicrobial biofilms, bacterial EPS can bind and sequester fluconazole, thereby reducing the uptake and intracellular transport of the drug and enhancing *C. albicans* tolerance to this widely used azole drug [55]. In polymicrobial *C. albicans*–*S. gordonii* biofilms, the biofilm matrix components produced by *C. albicans* protected *S. gordonii* against antimicrobial treatment with clindamycin [56]. The polymicrobial biofilm architecture and composition of the biofilm matrix can be modified in several ways, thereby greatly influencing the drug susceptibility of the polymicrobial community [57]. Further research into the biofilm matrix components is crucial for the development of improved treatment strategies with a better penetration into the biofilm structure.

### 2.3. Quorum Sensing: Communication is the Key

Communication within microbial communities, referred to as quorum sensing (QS), is mediated by the secretion of small metabolites and plays an important role in biofilm formation. However, QS might play an even bigger role in polymicrobial biofilms [58]. In *C. albicans*–*P. aeruginosa* mixed biofilms, the QS molecule 3-oxo-C12 homoserine lactone was found to induce *C. albicans* resistance to fluconazole [59]. This is accomplished by facilitation of the ergosterol synthesis which is the target of fluconazole. In addition, upregulation of drug efflux and maintenance of cell wall integrity further contribute to this phenomenon. Other studies, however, indicated that this QS molecule is able to inhibit *C. albicans* filamentation without affecting fungal growth rates [60]. Interestingly, this QS molecule structurally resembles an important *C. albicans* QS molecule, farnesol, which also inhibits the morphological shift from yeast to hyphae [59,61]. In *C. albicans*–*S. aureus* mixed biofilms, farnesol is partly responsible for the enhanced tolerance of *S. aureus* to vancomycin, by inducing oxidative stress which in turn triggers the upregulation of drug efflux pumps [62]. In contrast, mixed biofilm formation of *C. albicans* with *S. mutans* was reduced in the presence of farnesol [63]. Farnesol treatment reduced the total biomass, metabolic activity, and cell viability. In addition, the treatment of *C. albicans*–*S. aureus* biofilms with a combination of farnesol and gentamicin enhanced the antimicrobial efficacy of gentamicin, demonstrating the synergy between these compounds [64]. A second important QS molecule in *C. albicans* is tyrosol, which stimulates *C. albicans* hyphal formation [65]. Similar to farnesol, an antibiofilm effect was observed for tyrosol against mixed biofilms consisting of *C. albicans*, *C. glabrata*, and *S. mutans* on acrylic resin and hydroxyapatite surfaces [66]. Farnesol and tyrosol were also investigated in the context of *C. albicans*–*P. aeruginosa* biofilms. Tyrosol was found to exert antibacterial activity against *P. aeruginosa* by inhibiting bacterial growth and inhibiting the production of virulence factors hemolysin and protease [67]. However, there was no effect on the antibiotic resistance of *P. aeruginosa.* Farnesol inhibited bacterial growth and hemolysin production to a lesser extent. In the case of *S. gordonii*, autoinducer 2 (AI-2), a group of molecules that promotes cross-communication between bacteria, was found to enhance *C. albicans* hyphal formation during polymicrobial biofilm growth [44].

Prostaglandin E_2_, produced by *C. albicans*, is an oxygenated metabolite of arachidonic acid which is upregulated during biofilm formation. Prostaglandin E_2_ functions as a regulator of host immune responses by stimulating the activation, maturation, and migration of immune-related host cells [68]. This molecule was found to stimulate growth and biofilm formation of *S. aureus* in *C. albicans*–*S. aureus* mixed biofilms [69]. The accessory gene regulator (*agr*) QS system in *S. aureus* controls the expression of different virulence factors including α-toxin, which mediates hemolytic activity and inflammasome activation, and reduces macrophages phagocytic killing. Significantly elevated levels of both Prostaglandin E_2_ and α-toxin were observed during *C. albicans*–*S. aureus* co-infection in a mouse model of intraabdominal infection, while there was no difference in bacterial burden [70]. Further research should reveal the role of interspecies communication in fungal–bacterial biofilms and the possibility to target these QS systems in the treatment of co-infections. A major advantage of interfering with QS mechanisms is that it will control virulence and biofilm formation without causing a selective pressure.

## 3. Novel Treatment Strategies

Polymicrobial biofilms are usually more tolerant compared to their corresponding single species biofilms thereby limiting the possibility to use conventional drugs. However, voriconazole, a second-generation antifungal drug, was found to inhibit the interaction between *C. albicans* and *Actinomyces viscosus*, a Gram-positive bacterium associated with root caries, by inhibiting the ergosterol biosynthesis pathway [71]. In addition, the antifungal and antibacterial agents caspofungin and polymyxin B were able to reduce both the cell viability and total biomass of mixed biofilms consisting of *P. aeruginosa* with *C. albicans*, *C. parapsilosis*, *C. tropicalis* or *C. glabrata* [72]. The treatment of polymicrobial infections by using a combination of antibacterial and antifungal drugs is generally associated with a poor efficacy; however, combination therapy can also result in desirable outcomes. For example, in an intraabdominal catheter infection model in mice, it was found that anidulafungin acts synergistically with tigecycline against *C. albicans*–*S. aureus* mixed biofilms [73]. Similarly, an antimicrobial lock solution containing micafungin, doxycycline, and ethanol, used to sterilize infected catheters or medical devices, inhibited *C. albicans*–*S. aureus* mixed biofilms [74]. In addition, povidone iodine enhances the efficacy of fluconazole against mixed *C. albicans*–*S. mutans* biofilms by functioning as an EPS inhibitor through inhibition of α-glucan synthesis [55]. However, alternative approaches are being explored to treat fungal–bacterial biofilm infections and are discussed below and summarized in Table 1.

### 3.1. Antimicrobial Peptides

Antimicrobial peptides (AMPs) are short positively-charged molecules with an amphipathic character which are being explored as promising antimicrobial and antibiofilm compounds [102]. In nature, AMPs belong to the innate immunity of organisms to react against various pathogens and are, therefore, promising candidates in the development of new antibiotics. The main advantages are broad-spectrum activity, relatively low toxicity, rapid mechanism of action, and low probability of antimicrobial resistance. Several studies showed that AMPs can target bacteria and fungi by interacting with microbial membranes and disrupting the physical integrity [103,104]. However, the potential to use AMPs against polymicrobial infections has only been explored recently.

The membranotropic peptide gH625 (HGLASTLTRWAHYNALIRAF), derived from the herpes simplex virus type I, is a twenty-residue peptide which was shown to interact with and destroy membrane bilayers [105]. Later, this peptide was adapted by adding a sequence of lysine residues (gH625-GCGKKKK) which promotes both the interaction with negative charges of bacterial membranes and crossing of membrane bilayers and, therefore, interferes with biofilm formation [106]. The activity of the native and adapted peptide was assessed against monomicrobial and polymicrobial biofilms of *C. tropicalis*, *S. aureus*, and *Serratia marcescens* grown on silicone platelets. Although gH625 was not able to prevent *C. tropicalis* biofilm formation, the modified peptide displayed 90% inhibition. Similarly, for polymicrobial biofilms, the modified peptide showed a greater inhibition on biofilm formation compared to the native peptide. Both peptides appeared to be potent in the eradication of mature polymicrobial biofilms, likely due to the cell-penetrating capacity [75]. Recently, the antibiofilm activity of an analogue of the gH625 peptide, gH625-M (gH625-GGGKKKK), was evaluated against polymicrobial *C. albicans–Klebsiella pneumoniae* biofilms in vitro and in vivo [76]. Subminimal inhibitory concentrations of gH625-M were able to inhibit the formation of both mono- and polymicrobial in vitro biofilms, as well as to eradicate them. Furthermore, the in vivo antimicrobial activity of gH625-M was evaluated for the first time using *Galeria mellonella* larvae infected with *C. albicans* and *K. pneumoniae* isolates. The mortality of larvae co-infected with *C. albicans* and *K. pneumoniae* was slightly higher compared to single species infections. gH625-M was nontoxic for the larvae and the administration of gH625-M after and, especially, before infection greatly improved larvae survival rates. In addition, gH625-M treatment before infection significantly reduced the expression of *C. albicans* biofilm-associated genes *HWP1* and *ALS3* [76].

Unfortunately, the high production cost of AMPs and the chemical instability and pharmacokinetic properties hampers the development of AMP-based treatments. Therefore, alternative studies are using the characteristics of AMPs as an inspiration to develop small molecules which target bacterial or fungal membranes. In this regard, cholic acid was recently investigated for its activity against bacterial, fungal, and polymicrobial biofilms [77]. Cholic acid is a naturally-occurring bile acid with an amphiphilic nature. By adapting this molecule to better resemble the amphiphilic and cationic characteristics of AMPs, nonpeptide compounds referred to as “ceragenins” were developed [107]. Subsequently, cholic acid–peptide conjugates (CAPs), in which dipeptide units are conjugated on a cholic acid backbone, were shown to effectively interact with lipopolysaccharides and the valine–glycine dipeptide-derived CAP 3 was shown to be the most effective against Gram-negative bacteria [108]. Recently, CAP 3 was also tested against *C. albicans*, *S. aureus*, and polymicrobial biofilms [77]. Treatment with CAP 3 degraded preformed polymicrobial biofilms, and the formation of polymicrobial biofilms on CAP 3-coated catheters was prevented. In addition, CAP 3 was shown to be effective in polymicrobial wound and catheter infection models in mice. In a wound infection model, mice treated three times daily with CAP 3 showed approximately 2-log decreases in both *S. aureus* and *C. albicans* colonies. In a catheter infection model, CAP 3-coated catheters showed significant reductions in both bacterial and fungal loads [77].

Furthermore, AMP-mimicking synthetic polymers are under investigation for their anti-polymicrobial biofilm properties. These molecules have some considerable advantages as they are cheap and easy to produce and they can be chemically manipulated and integrated into drug delivery systems [109]. In this regard, guanylated polymethacrylates were synthesized, which are random copolymers of 2-guanidinoethyl methacrylate and methyl methacrylate, respectively, mimicking the cationic amino acid arginine and the lipophilic amino acid alanine. These molecules were recently shown to eradicate *C. albicans*–*S. aureus* polymicrobial biofilms in vitro [78]. Interestingly, when comparing the use of guanylated polymethacrylates with the use of different antimicrobial combinations, guanylated polymethacrylates were much more effective, with killing rates of >94% for *S. aureus* and >80% for *C. albicans* in the polymicrobial biofilm.

Finally, an ex vivo porcine wound biofilm model was developed to study the treatment of *P. aeruginosa*, *S. aureus*, and *C. albicans* polymicrobial biofilms [79]. As a treatment strategy, the natural peptide epsilon-poly-L-lysine was incorporated in a chitosan hydrogel and the antibiofilm efficiency was evaluated against mature polymicrobial biofilms ex vivo. The hydrogel was especially effective against *P. aeruginosa* when applied ex vivo to 24 h old polymicrobial wound biofilms; however, the activity was lower against *S. aureus*, and the hydrogel was ineffective against *C. albicans*. When the hydrogel was topically applied within 5 h after inoculation for 2 to 3 days, the biofilm thickness was reduced by at least 96% compared to untreated biofilms. Likewise, topical application of the hydrogel for 24 h to mature biofilms at 24 and 48 h after inoculation reduced biofilm thickness by >70%.

### 3.2. Quorum Quenchers

As described earlier, QS is a crucial process in polymicrobial biofilm formation and is, therefore, an interesting target for the development of novel treatment strategies. Interference in the process of QS, for example by inhibiting the synthesis or interaction of QS molecules, is referred to as quorum quenching. Quorum-quenching molecules have already been used as novel therapeutic agents combating bacterial or fungal biofilms; however, the possibility to use them against polymicrobial biofilms should be investigated further. Derivatives of thiazolidinediones, which are inhibitors of QS in the Gram-negative bacterium *Vibrio harveyi,* were developed as antibiofilm agents against *C. albicans* [110,111]. One of these derivatives, thiazolidinedione-8, profoundly affected *C. albicans* biofilm formation and was able to destroy preformed biofilms [111,112]. To further increase its therapeutic potential as a drug against oral candidiasis, thiazolidinedione-8 was incorporated in a sustained-release membrane, thereby allowing a prolonged release of the drug in the oral cavity [113]. This inhibited *C. albicans* in vitro biofilm formation in a time-dependent manner and significantly eradicated mature biofilms. Additionally, it was shown that thiazolidinedione-8 decreases the biomass of *C. albicans*–*S. mutans* mixed biofilms in vitro, although no effect was observed against *S. mutans*, resulting in a favorable effect for *S. mutans* [80]. This was further investigated using a model system with constant flow conditions to mimic *C. albicans*–*S. mutans* mixed biofilm formation in the oral cavity [81]. Thiazolidinedione-8 incorporated in a sustained-release membrane allowed better penetration of the compound in the mixed species biofilm, thereby affecting both pathogens by decreasing the biofilm metabolic activity and the production of EPS and altering the morphology of both pathogens.

Other quorum-quenching molecules were identified through functional screening of metagenomic large insert libraries. Two naturally-occurring quorum-quenching enzymes (QQ-5 and QQ-7) were found to potentially interfere with *C. albicans* and *S. epidermidis* biofilm formation [82]. QQ-5 and QQ-7 were able to prevent in vitro *C. albicans* biofilm formation by inhibiting the process of morphogenesis. Inhibition of *S. epidermidis* biofilm formation was likely due to the induction of a repressor for polysaccharide intracellular adhesin (PIA), which plays a key role in *S. epidermidis* biofilm formation. These findings highlight the potential of using such quorum-quenching molecules in targeting fungal–bacterial biofilms. Non antibiotic pharmacological agents can be assessed for QS inhibition to identify new quorum-quenching molecules [114]. This drug development strategy is referred to as drug repurposing and offers some valuable advantages, as the toxicity and pharmacokinetic profile of these drugs are already established.

### 3.3. Plant-Derived Components

Plants protect themselves from microbial infections by the production of different types of antimicrobial molecules, including essential oils (EOs) and essential oil components. A large number of studies have been published where such compounds were tested for their antimicrobial or antibiofilm activity against either bacterial or fungal species [115,116,117,118]. More recently, several such molecules have also been tested in mixed fungal–bacterial settings. An example is citral, that shows strong activity against MRSA, where it targets biofilm specific pathways and also shows activity against different *Candida* species [119,120,121]. Recently, citral, as well as lemongrass, which is rich in citral and nepodin, were shown to have strong activity against dual-species *C. albicans*–*S. aureus* biofilms [83,84]. Citral resulted in lower biofilm biomass as well as in the number of viable cells of both species. The authors also showed that citral was interfering with the expression of *Candida* adhesins, as well as downregulating the expression of genes involved in the production of QS molecules, peptidoglycan, and fatty acids in *S. aureus* [84]. Nepodin, isolated from *Rumex crispus*, was shown to reduce the expression of hypha-specific genes and upregulated the expression of several transport genes [83]. Whereas citral was only tested in vitro, nepodin also affected *C. albicans* virulence in a nematode infection model system, but dual-species infections were not performed.

Citrus EOs obtained from pompia and grapefruit, and their major compound, limonene, were tested for their activity against mixed *P. aeruginosa–Aspergillus fumigatus* or *Scedosporium apiospermum* biofilms, which are relevant species in cystic fibrosis patients [85]. After a 24-h treatment, biofilms composed of *P. aeruginosa* and *A. fumigatus* were completely inhibited, and biofilms composed of *P. aeruginosa* and *S. apiospermum* were significantly reduced. Interestingly, these EOs showed the ability to interfere with the QS system of *P. aerugonisa* and to cause *Candida* membrane damage.

Eugenol, the major component of EOs from the *Syzygium aromaticum* plant (clove), was tested for its antibiofilm activity in the management of oral diseases [86]. A concentration-dependent antibiofilm activity was observed against polymicrobial *C. albicans–S. mutans* biofilms with a reduction in biofilm formation of 52% at a subminimal inhibitory concentration of eugenol. In addition, the cell viability was significantly reduced, and cell membranes and matrix structures were disrupted. Moreover, eugenol is nontoxic and safe for human use and should, therefore, be further evaluated in vivo. A combination therapy of conventional drugs with geranium, citronella, and clove EOs was also assessed against *C. albicans–S. aureus* biofilms [122]. Treatment of these preformed polymicrobial biofilms with fluconazole or mupirocin in combination with clove oil was the most effective and resulted in a 10-fold and 4-fold increase in antibiofilm activity of fluconazole and mupirocin, respectively.

Curcumin is a polyphenol isolated from the spice *Curcuma longa* (turmeric) with a wide range of pharmacological activities, including antibacterial and antifungal effects [123]. However, the antibiofilm activity against polymicrobial *C. albicans–S. aureus* biofilms was only explored recently. Curcumin significantly reduced the formation of both mono- and polymicrobial biofilms and was effective against preformed biofilms, although the effect was smaller [87]. Interestingly, the effect was enhanced when curcumin was combined with 2-aminobenzimidazole. The antimicrobial effect of curcumin was also evaluated against mixed biofilms of *C. albicans* with *Acinetobacter baumannii*, a Gram-negative bacterium responsible for a variety of nosocomial infections [88]. Curcumin was able to reduce mixed biofilm formation by >85% and reduced *A. baumannii* virulence in a *C. elegans* model without exhibiting toxicity; however, the in vivo activity against mixed biofilms was not assessed. Unfortunately, the therapeutic application of curcumin is hindered due to its instability and low bioavailability [124].

The phenolic essential oil component carvacrol has broad spectrum antimicrobial properties and was already proved to be effective against oral and vaginal candidiasis in vivo [125,126]. However, carvacrol has a poor solubility and high volatility, which limits the application potential. Therefore, drug delivery systems using electrospun nanofibers were explored to improve the stability and prolong the activity [89]. Carvacrol-containing polylactic acid electrospun nanofibrous membranes were evaluated against *C. albicans–S. aureus* biofilms. Carvacol was gradually released from the membrane which ensured an antimicrobial activity up to 144 h. Both the formation of polymicrobial biofilms and metabolic activity, vitality, and biomass of preformed biofilms were reduced, indicating the potential of this compound for skin and wound infections [89]. Finally, a stem extract from the small *Rhamnus prinoides* tree (gesho) was able to inhibit *S. mutans–C. albicans* biofilm formation by 98%, likely by reducing the EPS production [90].

### 3.4. Photodynamic Therapy

Photodynamic therapy involves the combination of visible light with nontoxic dyes, called photosensitizers, to kill microbial cells including bacteria, fungi, and viruses [127]. The photosensitizers are irradiated by light of the wavelength, which they are able to absorb to generate oxidized products and singlet oxygen capable of damaging essential cell components. Photodynamic therapy appears to be very promising within various subspecialties of dentistry by offering a minimally-invasive antimicrobial treatment modality. In addition, several photosensitizers are already approved for safe use in dentistry and there is no threat of antibiotic resistance against photodynamic therapy [128].

By using erythrosine as a photosensitizer and a green light-emitting diode, significant microbial reduction of both *C. albicans* and *S. sanguinis* in a mixed in vitro biofilm was observed [91]. However, mixed biofilms of *C. albicans* with *S. sanguinis*, *S. mutans*, or *S. aureus* were still more resistant to photodynamic inactivation compared to single species biofilms, likely due to the more complex biofilm matrix [91,129,130]. Erythrosine is already approved for use in dentistry and was used because of its nontoxic effect to the host. In addition, green LED light has some noteworthy advantages compared to other lasers, such as the cost, smaller size, and broader emission bands.

Furthermore, acrylic resins doped with *Undaria pinnatifida*, a brown seaweed microalga which contains the natural photoactive pigment chlorophyll-a, were used as photosensitizers in combination with blue light to target multispecies biofilms [92]. This study showed a dose-dependent reduction of viable microbial cells in biofilms of *C. albicans*, *S. sanguinis*, *S. mutans*, and *Lactobacillus acidophilus* and inhibition of biofilm formation of all microorganisms by 99%.

Finally, photodynamic therapy was evaluated for the eradication of endodontic biofilms by using the photosensitizer Zn(II)chlorin e6 methyl ester, obtained from chlorophyll-a, in combination with red light [93]. In an in vitro model of endodontic *C. albicans*–*Enterococcus faecalis* biofilm, the photodynamic therapy was able to remove 60% of the biofilm mass and was, therefore, more efficient compared to other photosensitizers. In the endodontic field, a microbial infection in the root canal system caused by multispecies biofilms remains a common problem, which led to the emergence of photodynamic therapy in this field.

### 3.5. Carboxymethyl Chitosan

Chitosan is a natural polysaccharide polymer that has received increasing attention in the medical world because of its antimicrobial activity combined with a low cost, good accessibility, and low toxicity [131]. However, chitosan has a poor solubility in water, which has been resolved by carboxymethylation, yielding carboxymethyl chitosan (CMC). Chitosan is active on the cell surface of fungi and bacteria via electrostatic interactions, resulting in permeabilization of the cells. The antibiofilm effect of CMC was assessed against mixed fungal–bacterial biofilms developed on silicone plates, often used for the construction of medical devices and voice prostheses [94,95]. Mixed biofilms consisted of *C. albicans*, *C. tropicalis*, *S. epidermidis*, *Streptococcus salivarius*, *Rothia dentocariosa,* and *Lactobacillus gasseri*, all isolated from voice prostheses of laryngectomized patients. Treatment with CMC inhibited biofilm formation by approximately 73% and significantly decreased the biofilm metabolic activity. In addition, CMC inhibited *Candida* morphogenesis and adhesion of both fungi and bacteria. Furthermore, CMC could effectively inhibit both mono- and polymicrobial biofilms of *C. tropicalis* and *S. epidermidis* in microplates and on silicone surfaces [96]. Polymicrobial biofilms were inhibited by 56.2% and 54.7% on microtiter plates and on silicone surfaces, respectively. Although CMC itself displays antimicrobial activity, the effect of CMC can also be further enhanced with the use of various nanoparticles. In addition, further animal model studies are crucial for the evaluation of CMC in vivo.

### 3.6. Nanoparticles

A promising solution to overcome the limitations of conventional therapies is to employ the use of nanoparticles (NPs). Nanomaterials behavior is different from the bulk material, mainly due to their high surface area-to-volume ratio [132]. Consequently, nanostructures have mechanical, electrical, chemical, and magnetic properties that can be an advantage for drug delivery purposes. The small size of nanocarriers facilitates their penetration into the biofilm, ensuring the release of their antimicrobial contents locally. In addition, the small size allows efficient interaction with the microbial membrane and, eventually, with the nuclear content of the pathogen, thereby interfering with cellular processes [133]. The surface charge of the NPs also plays a role in the interaction with the biofilm, since positively-charged NPs bind to the negatively-charged EPS through electrostatic interactions, which allows them to reach deeper regions. Besides being a valuable tool as nanocarriers, NPs may also show intrinsic antimicrobial properties, such as silver and gold NPs [134]. These particles inhibit QS and generate reactive oxygen species (ROS), leading to cell damage [132,135]. Due to these unique characteristics, NPs have been explored to target polymicrobial biofilms (Table 2).

Several studies reported the use of NPs as drug delivery systems of natural compounds with antimicrobial properties. Taking advantage of the antimicrobial activity of curcumin, curcumin-loaded chitosan NPs were developed and tested against *S. aureus* and *C. albicans* mono- and polymicrobial biofilms [124]. The ability of these NPs to penetrate biofilms was evaluated, and NPs were observed both at the surface and deeper regions of the biofilm, which is likely a consequence of the NPs’ positive charge, promoting its interaction with negatively-charged biofilm components and microbial cells. Thus, the low diffusion of free curcumin into the biofilm is overcome by encapsulation into NPs [124]. Curcumin-loaded NPs were more effective against preformed biofilms compared to free curcumin, with a biofilm reduction of 84.36%. In a more complex approach, curcumin-loaded chitosan NPs were designed for antimicrobial photodynamic therapy, where the antimicrobial properties of the photosensitizer curcumin would be potentiated in the presence of light by ROS production [150,151]. The efficiency of anionic and cationic curcumin-loaded NPs was evaluated against mono-, dual-, and triple-species biofilms of MRSA, *S. mutans,* and *C. albicans*. For the anionic formulation, only *S. mutans* in mono-species biofilms was susceptible, possibly due to the repulsion between the NPs and the anionic components of the biofilm matrix [136]. Therefore, cetyltrimethylammonium bromide (CTAB) was further incorporated into the nanoformulation to produce cationic curcumin-loaded NPs. Both unloaded and curcumin-loaded cationic NPs showed a reduction of microbial viability in multi-species biofilms, even in the absence of light. This effect may be attributed to their positive charge, which promotes interactions with the microbial cell membrane [136]. Additionally, CTAB has been demonstrated to promote cell lysis of *S. aureus*, *E. faecalis,* and *Escherichia coli* [152]. Despite their potential, these cationic NPs are not safe systems for in vivo applications due to their cytotoxicity towards mammalian cells [136].

The previous studies showed the potential of nanocarriers to encapsulate antimicrobial agents. However, the biofilm matrix plays an important role in protecting microbial cells, thus, it is important to combine an approach that not only kills microbial cells but also disrupts matrix components [153]. In a combinatory strategy, cellobiose dehydrogenase (CDH) and deoxyribonuclease I (DNase I) were co-immobilized on chitosan NPs to target both microbial cells and the biofilm matrix [137]. CDH is widely used as an antimicrobial agent, while DNase I is able to degrade eDNA in the matrix of biofilms [154,155,156]. CDH–DNase NPs inhibited *C. albicans–S. aureus* mixed biofilm formation by 90.5% [137]. For preformed biofilms, the formulation showed a disruption higher than 80%. Both CDH and DNase I were also immobilized separately into NPs (CDH NPs and DNase NPs, respectively) [137]. As expected, DNase NPs had no effect on biofilm since it is not able to kill microbial cells, which allows biofilms to be constantly formed even after dispersal. However, CDH–DNase NPs showed a higher antibiofilm activity than CDH NPs, indicating a synergistic effect of DNase I by improving NPs penetration into the biofilm. Therefore, the developed nanosystem may present a safe therapeutical strategy to eradicate polymicrobial biofilms [137]. In another study, the synergy between chitosan NPs and olive oil in the treatment of oral biofilms was assessed [138]. For this purpose, an ex vivo premolar teeth model was used to grown mature mixed biofilms of *E. faecalis*, *S. mutans*, and *C. albicans*. The combinatory treatment of olive oil and chitosan NPs led to a 6-log reduction of viable cells in only two days, while chitosan NPs or olive oil alone needed at least one week to provide acceptable viability reduction [138].

Despite their advantages as nanocarriers, chitosan NPs alone have been investigated as a therapeutical strategy against polymicrobial biofilms. The in vitro efficiency of chitosan NPs was assessed against *C. albicans–S. mutans* biofilm formation. A decreasing trend of remaining biofilm biomass and a significant cell viability decrease with the increase of concentration of chitosan NPs was observed [139]. Besides chitosan, other biocompatible polymers have been used in nanosystems for antibiofilm purposes. Recently, alginate-based nanocarriers loaded with lipid extracts from cyanobacteria *Arthrospira platensis* were developed. The activity of the extracted lipids was assessed in dual-species *C. albicans–Cutibacterium acnes* biofilms. The lipid-loaded alginate NPs were successful in inhibiting biofilm growth and disrupting preformed single-species *C. albicans* biofilms, which was not verified for the free lipid extracts. However, limited efficiency was observed when these NPs were tested in dual-species biofilms [140].

Recently, NPs with magnetic properties were explored as nanocarriers to eradicate polymicrobial biofilms. These NPs generate ROS and have, therefore, intrinsic antimicrobial activity [157]. Besides, magnetic NPs are advantageous for a targeted delivery to a specific site, since it is possible to guide them using an external magnetic field [157]. Chlorhexidine (CHX) was immobilized onto the magnetic NPs surface to enhance its antimicrobial activity [141]. Both free CHX and CHX-loaded magnetic NPs were tested in the presence of human saliva for its efficiency against multi-species oral biofilms. It was observed that immobilized CHX had an increased ability to restrict biofilm growth compared to free CHX at the same concentration. Additionally, the immobilized CHX did not exhibit cytotoxic effects against human osteoblast cells, while the free agent induced significant toxic effects [141]. To improve its biocompatibility and stability, CHX-loaded magnetic NPs were coated with chitosan [142]. The developed particles showed a significant reduction in biofilm biomass of dual-species *C. albicans–S. mutans* biofilms. The in vitro efficiency of the formulation was also assessed against preformed biofilms. The nanocarrier and the free CDX showed a metabolic activity reduction of 94.4% and 89.7% against dual-species biofilms, respectively. Thus, CDX-loaded nanocarriers may promote a higher antibiofilm effect of the drug, while lowering its toxicity toward human cells [142]. Magnetic NPs coated with chitosan were also used to load miconazole, a drug with both antibacterial and antifungal properties [143,158]. This formulation was further tested on three representative interkingdom oral biofilms (caries, denture, and gingivitis) [143]. In all three models, nanocarriers containing miconazole significantly reduced the number of viable cells. Interestingly, the formulation also promoted changes in the predominance of the different species composing the biofilm, with bacterial cells showing higher susceptibility to the treatment. In addition, fewer hyphae were observed in the presence of the designed NPs, which suggests an inhibition of *C. albicans* hyphal form during treatment. Consequently, the support provided by hyphae to bacteria is partially lost, explaining the lower numbers of bacterial cells [143].

The potential of NPs for the development of antimicrobial surface coatings to avoid biofilm formation has been also highly explored in the past years. Mesoporous silica NPs functionalized with phenazine-1-carboxamide (PCN) were produced to coat silicone urethral catheters. The authors assessed the antibiofilm activity of PCN extracted from *P. aeruginosa* and hypothesized that this metabolite induces ROS accumulation and reduction of ergosterol content. The NPs showed a controlled release of PCN over a 40-h period, which contributed to an inhibition higher than 88% of *C. albicans–S. aureus* biofilms. In fact, PCN-loaded NPs were able to inhibit mixed biofilms at a very low concentration compared to unloaded PCN [145]. In a distinct approach to inhibit formation of *C. albicans–S. aureus* biofilms in catheters, silicone elastomers were functionalized with silver NPs. The silver NPs effectively prevented mixed biofilm formation; however, the study lacks cytotoxicity assessment of the developed coating, which may hinder in vivo applications [147]. Silver NPs were also reported as a surface coating to avoid formation of *C. albicans–P. aeruginosa* biofilms [148]. Polyethylene and silicon substrates functionalized with silver NPs efficiently inhibited biofilm formation in *P. aeruginosa* monocultures, however this effect was not verified for mixed biofilms. It is believed that limited efficiency of silver NPs in coatings may be a consequence of a fast silver release from the NPs and moisture intake [148,149]. To overcome this drawback, a coating composed of silver NPs and polyamide was developed to decrease microbial adherence to endotracheal tubes [149].

### 3.7. Probiotics

Another strategy to target polymicrobial fungal–bacterial biofilms is the use of probiotics, which are defined as live microorganisms with beneficial effects on health when administered in adequate amounts. In patients with Crohn’s disease, an inflammatory disease of the bowel, an increase in the abundance of *C. albicans*, *C. tropicalis*, *E. coli,* and *S. marcescens* was observed [159]. Therefore, probiotics which target these pathogenic microorganisms and support beneficial microorganisms are being explored in the setting of Crohn’s disease [160]. A novel probiotic formulation was developed consisting of *Saccharomyces boulardii*, *L. acidophilus*, *Lactobacillus rhamnosus,* and *Bifidobacterium breve* combined with amylase for its antibiofilm activity. This probiotic formulation possessed antibiofilm activity against polymicrobial biofilms grown on silicone elastomer discs consisting of *C. albicans* or *C. tropicalis* in combination with *E. coli* and *S. marcenscens* by reducing the polymicrobial biofilm matrix and thickness and inhibiting *Candida* hyphal formation [99]. Polymicrobial biofilm growth of *C. albicans*, *C. tropicalis*, *S. salivarius*, *R. dentocariosa,* and *S. epidermidis* on silicone medical devices, such as voice prostheses, increases the risk of infection and limits the lifetime of the prosthesis. Therefore, the supernatant of probiotic *Lactobacillus,* containing exometabolites with antimicrobial activity, was tested as a treatment for these fungal–bacterial biofilms on silicone material [100]. This *Lactobacilli* supernatant was able to inhibit the adhesion and biofilm formation and reduce the polymicrobial biofilm metabolic activity. In addition, this probiotic treatment was able to inhibit *Candida* hyphal formation. The beneficial effects of probiotics were also explored in the context of early childhood caries, of which *S. mutans* is the main etiological agent [101]. Coexistence of *S. mutans* with *C. albicans* appears to be involved in dental caries progression and recurrence. A probiotic containing *Lactobacillus salivarius* was shown to decrease the biofilm mass and inhibit *Candida* hyphal formation in dual-species biofilms of *S. mutans* with *C. albicans*, thereby weakening its pathogenic potential.

### 3.8. Other Treatment Strategies

A series of compounds based on a 2-aminoimidazole scaffold were constructed and tested for their antibiofilm capacity against *Salmonella* Typhimurium and *P. aeruginosa* [161]. The nontoxic N1- and 2N-substituted 5-acryl-2-aminoimidazoles compounds were screened for antibiofilm activity against polymicrobial biofilms as well [162]. These compounds showed a strong activity against polymicrobial biofilm formation by Gram-positive bacteria and *C. albicans*; however, there was no activity against Gram-negative bacteria.

Dentures are an ideal surface for polymicrobial biofilm formation and can, therefore, lead to denture-related stomatitis. The possibility to reduce biofilm formation on polymethyl methacrylate denture material by incorporating antimicrobials into the material is being investigated [163]. The antimicrobial properties against mixed biofilms consisting of *C. albicans*, *Lactobacillus casei,* and *S. mutans* of a novel fluoride-releasing material were compared with a non-fluoridated copolymer. This study showed that the fluoride release significantly reduced the cell densities of all three species, highlighting the potential to include fluoride in dentures to control biofilm growth and subsequent diseases.

Bacteriophages are viruses which are able to infect bacteria and reproduce inside. The use of bacteriophages in phage therapy is a promising strategy against pathogenic bacterial infections and was shown to be effective against biofilm-related infections as well [164]. A few studies also investigated the activity of phages and the synergistic activity of phages with antibiotics against bacterial polymicrobial biofilms [165,166]. In addition, the possibilities of phage therapy are being explored in the context of nonbacterial infections [167]. In this context, it was shown that a bacteriophage of *P. aeruginosa* is not only able to inhibit *P. aeruginosa* biofilms, but also *A. fumigatus* and *C. albicans* biofilms [168]. Both biofilm formation and preformed biofilms of *C. albicans* were inhibited, likely by iron denial [169]. The drug repurposing strategy to use such phages for the treatment of mixed fungal–bacterial infections has not yet been explored but could have a lot of potential.

## 4. Conclusions

Polymicrobial diseases are increasingly being recognized in clinical settings and effective treatment strategies are lacking. Therefore, understanding the molecular mechanisms of the interactions between fungal and bacterial species in different niches of the host are of great importance. However, fungal–bacterial infections have been studied most extensively in the context of the oral microbial flora, and there is still a lack of research on these interactions in other anatomical areas or pathologies.

Although several promising techniques are highlighted here, the development of innovative therapeutic strategies to prevent interkingdom biofilms is still in its infancy. Several studies indicated the in vitro potential of antimicrobial peptides, plant-derived components, quorum quenchers, and probiotics, however, there remain hurdles to be overcome. The in vivo evaluation of the activity of these new therapeutic strategies in animal models and the assessment of possible toxicity are crucial but currently rather limited. Nanotechnology has also been highlighted as a promising tool to fight polymicrobial biofilms. Nanoparticles can be used as drug delivery systems to improve their antimicrobial effect, while decreasing their toxicity toward human cells. Besides, due to their small size, nanoparticles can reach deeper layers of the biofilm structure, leading to a higher concentration of antimicrobial agents in these regions. The intrinsic antimicrobial properties of nanoparticles have been also explored to overcome resistance phenomena and to avoid biofilm formation on medically relevant substrates. Although the reported studies showed in vitro efficacy against mixed biofilms, in vivo studies to evaluate the antibiofilm efficacy are lacking.

On a concluding note, the clinical need for the development of new strategies to target fungal–bacterial biofilm infections is high, and requires a good understanding of the interactions within these biofilms. In addition, some important properties of new drugs including broad-spectrum activity, no toxicity, and availability for oral administration should always be taken into account during the development [170]. Therefore, new components only rarely succeed from the preclinical to the clinical phase, and the quest for novel therapeutic strategies remains extremely challenging.

## Figures and Tables

**Table 1 microorganisms-09-00412-t001:** Summary of alternative treatment strategies for fungal–bacterial biofilms.

Treatment	Strengths	Limitations	Examples	Biofilm Target	Ref.
Antimicrobial peptides	Broad-spectrum activity Low toxicity Low probability of resistance Rapid Efficient	Chemical instability High production cost Pharmacokinetic properties	gH625 analogues	*C. tropicalis–S. aureus–S. marcescens* *C. albicans–K. pneumoniae*	[75,76]
			cholic acid-peptide conjugates	*C. albicans–S. aureus*	[77]
			guanylated polymethacrylates	*C. albicans–S. aureus*	[78]
			ε-poly-L-lysine in chitosan hydrogel	*P. aeruginosa–S. aureus–C. albicans*	[79]
Quorum quenchers	Selective pressure only under QS conditions Low probability of resistance	May disturb microbiota homeostasis May cause enhanced virulence	thiazolidinedione-8	*C. albicans–S. mutans*	[80,81]
			QQ-5 and QQ-7	*C. albicans–S. epidermidis*	[82]
Plant-derived components	Wide variety of pharmaceutical and biological activities Low toxicity	High volatility Low stability Low bioavailability Small scale production	citral and nepodine	*C. albicans–S. aureus*	[83,84]
			citrus EOs and limonene	*P. aeruginosa–A. fumigatus or S. apiospermum*	[85]
			eugenol	*C. albicans–S. mutans*	[86]
			curcumin	*C. albicans–S. aureus* *C. albicans–A. baumannii*	[87,88]
			carvacrol	*C. albicans–S. aureus*	[89]
			*Rhamnus prinoides* stem extract	*C. albicans–S. mutans*	[90]
Photodynamic therapy	Broad-spectrum activity No toxicity Low probability of resistance	Limited effect against biofilms in vitro studies rarely translate into animal models	erythrosine—green light	*C. albicans–S. sanguinis*	[91]
			acrylic resins doped with *Undaria pinnatifida*—blue light	*C. albicans–S. sanguinis–S. mutans–L. acidophilus*	[92]
			Zn(II)chlorin e6 methyl ester—red light	*C. albicans–E. faecalis*	[93]
Chitosan	No toxicity BiodegradableLow cost Good accessibility Low immunogenicity	Poor solubility in water	carboxymethyl chitosan	*C. albicans–C. tropicalis–S. epidermidis–S. salivarius–R. dentocariosa–L. gasseri*	[94,95]
				*C. tropicalis–S. epidermidis*	[96]
Nanoparticles	Enhanced bioavailability of loaded drugs Targeted delivery Easier penetration inside biofilm Protection of drugs from external environment	Possible toxicity to mammalian cells Unknown processes of in vivo metabolism clearanceLong-term toxicity Difficult scale-up High-cost	polymeric NPs magnetic NPs mesoporous silica NPs silver NPs	cf. Table 2	[97,98]
Probiotics	Restores and maintains the balance of microbiota Good accessibility Easy to use	Limited survival of viable probiotic cells Lack of clinical studies and mode-of-action studies	*S. boulardii–L. acidophilus–L. rhamnosus–B. breve* with amylase	*C. albicans* or *C. tropicalis–E. coli–S. marcenscens*	[99]
			supernatant probiotic *Lactobacillus*	*C. albicans–C. tropicalis–S. salivarius–R. dentocariosa–S. epidermidis*	[100]
			*L. salivarius*	*C. albicans–S. mutans*	[101]

**Table 2 microorganisms-09-00412-t002:** Summary of nanoparticles developed as a therapeutic approach against interkingdom biofilms.

Nanoparticles	Formulation and Associated Compounds	Applications	Mechanism of Action	Ref.
Polymeric NPs	chitosan sodiumtripolyphosphate curcumin	Medical devices-associated infections *C. albicans*–*S. aureus*	Increase bioavailability Enhanced antimicrobial activity	[124]
	polylactic acid, dextran sulfate, CTAB (cationic) curcumin	Oral biofilms *S. mutans*–*C. albicans*–MRSA	Increase bioavailability Improve water solubility Decrease cytotoxicity Improve photodynamic effect	[136]
	chitosan sodiumtriphosphate Functionalization: CDH, DNase I	Medical devices-associated infections *C. albicans*–*S. aureus*	Disrupt EPS Enhanced antimicrobial activity Improve physical stability	[137]
	chitosan NPs ozonated olive oil	Endodontic infections *E. faecalis*–*S. mutans*–*C. albicans*	Synergy between ozonated olive oil and chitosan NPs	[138]
	chitosan tripolyphosphate	Early childhood caries *S. mutans*–*C. albicans*	Enhanced antimicrobial activity	[139]
	alginate, copper Solvents: EtOAc, DMC *A. platensis* lipid extract	*C. albicans*–*C. acnes*	Increase bioavailability	[140]
Magnetic NPs	iron chloride salts ammonium hydroxide Coating: aminosilane chlorhexidine	Oral biofilms *C. albicans–*MRSA–*P. aeruginosa*–*E. faecalis*	Decrease effective dosage Enhanced bioavailability Enhanced biocompatibility Enhanced antimicrobial activity	[141]
	iron oxide NPs Coating: chitosan chlorhexidine	Oral biofilms *C. albicans*–*S. mutans*	Decrease effective dosage Enhanced bioavailability	[142]
	iron oxide NPs Coating: chitosan miconazole	Caries, dentures, gingivitis *C. albicans*–*F. nucleatum*–*F. nucleatum vincentii*–*V. dispar*–*A. naeslundii*–Streptococci–*L. zeae*–*L. casei*–*R. dentocariosa*	Increase bioavailability	[143,144]
Mesoporous silica NPs	CTAB, tetraethoxysilane Functionalization: phenazine-1-carboxamide	Infections associated to urethral catheters *C. albicans*–*S. aureus*	Increase bioavailability Lower effective dosage Controlled drug release	[145]
Silver NPs	silver nitrate	Infections associated with catheters *C. albicans*–MRSA	Enhanced antimicrobial activity Prevent surface colonization	[146,147]
	branched polyethylenimine silver nitrate	Biofilm-based nosocomial infections *C. albicans*–*P. aeruginosa*–*S. aureus*	Prevent surface colonization	[148]
	polyamide, silver nitrate *Eucalyptus citriodora* leaves extract	Ventilator-associated pneumonia *C. albicans*–*P. aeruginosa*–*S. aureus*	Prolonged antimicrobial activity	[149]

NPs: nanoparticles; CTAB: cetyltrimethylammonium bromide; MRSA: methicillin-resistant *S. aureus*; CDH: cellobiose dehydrogenase; DNaseI: deoxyribonuclease I; EPS: extracellular polysaccharides; EtOAc: ethyl acetate; DMC: dimethyl carbonate.

## Data Availability

The data presented in this study are available upon request from the corresponding author.

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
