# Peer review of "Microbial Interkingdom Biofilms and the Quest for Novel Therapeutic Strategies"

_microorganisms, 2021, doi:10.3390/microorganisms9020412_

Round 1

Reviewer 1 Report

Dear authors,

In the review entitled “Microbial interkingdom biofilms and the quest for novel therapeutic strategies”, the authors have done an extensive revision about the main mechanisms involved in polymicrobial drug tolerance and alternative treatment strategies for fungal-bacterial biofilms.

All the paper is well-written and well-structured allowing a fluent and interesting reading.

1) Concerning the novel treatment strategies, I would like to know the opinion of the authors about “Antisense therapy”, “Page therapy” and “Phage-Antibiotic Combination” against fungal/bacterial biofilms/multispecies biofilms. Also, I would like to know if it is important talk about these therapeutic strategies in this paper?

2) The Table 1 is an excellent summary of alternative treatment strategies; however, it would be helpful an image for each treatment strategy, showing the mechanisms of action of each one.

Author Response

With this communication, we would like to clarify how we addressed the comments presented by the two reviewers to our first submission. We are pleased that we were able to address the comments of the reviewers and we are convinced that this ameliorated our manuscript to a great extent.

Reviewer 1:

1) Concerning the novel treatment strategies, I would like to know the opinion of the authors about “Antisense therapy”, “Page therapy” and “Phage-Antibiotic Combination” against fungal/bacterial biofilms/multispecies biofilms. Also, I would like to know if it is important talk about these therapeutic strategies in this paper?

Phage therapy could be potentially interesting in the treatment of fungal-bacterial biofilms as it is already shown to be effective against biofilm infections and in addition it is shown to be effective against certain fungal species as well. Unfortunately, no studies have focused on the use of phage therapy or a phage-antibiotic combination to specifically target mixed fungal-bacterial biofilms so far. However, since it still poses a potential interesting treatment strategy, we have added a small paragraph on this topic in section 3.8 (line 791). Regarding antisense-therapy, the literature is scarcer and there is no available literature where they test a combination of fungal and bacterial pathogens. In addition, there is not yet any literature which suggests a possible use of antisense therapy for polymicrobial fungal-bacterial biofilms and we have therefore not included it in this manuscript.

2) The Table 1 is an excellent summary of alternative treatment strategies; however, it would be helpful an image for each treatment strategy, showing the mechanisms of action of each one.

We agree that it might be interesting to have a figure, although, we doubt whether this would be extra informative in addition to the table. Instead of creating an additional figure, we have adapted table 1 and to have an easier to read overview, like reviewer 2 suggested.

Reviewer 2 Report

General: the paper is generally well-written, but the English use in some parts of the paper is a bit sloppy. the authors need to look into the use of definite and indefinite articles , anf the Oxford comma

  1. Introduction: some sentences feel redundant, the text could be more condensed and crisp.

L24: Please rephase the sentence.

L26: „are highly associated with human health and disease.” please rephase and be more precise.

L28: may be located

L34-L39: please say a few words regarding the intrinsic resistance of various Candida species other than C. albicans, as this is an important differentiating factor in clinical practice.

„Gram-positive” and „Gram-negative” is the correct way of writing, use it throughout the paper!

L47: please consider including the following reference:

https://pubmed.ncbi.nlm.nih.gov/31052511/

L49: what do you mean by increased virulence?

Section 2.

In some parts of this section, the authors are merely recounting the results of other authors without putting them in context or sythesizing them/generating new ideas.

The correct way to write is „gentamicin” (cf. tobramycin, this depends on the originator)

Section 3.

In this section, please include additional discussion on certain areas in light of the following references:

https://pubmed.ncbi.nlm.nih.gov/32392793/

https://www.ncbi.nlm.nih.gov/pmc/articles/PMC6429336/

Table 1. is a bit disorganized and hard to read, the authors should make an effort to improve clarity

3.2. please discuss the possibility of using the drug repurposing strategy to find novel QS-inhibitors:

https://pubmed.ncbi.nlm.nih.gov/31861228/

3.8. the authors should expand on more relevant strategies. e.g, if biofilm-ihibition in concerned, bacteriophages were shown to be effective

Other: there was an unreasonable amount of emphasis on oral infections and the oral microflora. Would it be relevant to also explore other anatomical areas/infectious pathologies where polymicrobial infections are relevant?

Author Response

With this communication, we would like to clarify how we addressed the comments presented by the two reviewers to our first submission. We are pleased that we were able to address the comments of the reviewers and we are convinced that this ameliorated our manuscript to a great extent.

Reviewer 2:

General: the paper is generally well-written, but the English use in some parts of the paper is a bit sloppy. the authors need to look into the use of definite and indefinite articles, and the Oxford comma

We have checked the manuscript for incorrect use of definite and indefinite articles and the Oxford comma and corrected where necessary. In addition, we have gone through the entire manuscript to correct for English language and style.

1) Introduction: some sentences feel redundant, the text could be more condensed and crisp.

L24: Please rephase the sentence.

L26: „are highly associated with human health and disease.” please rephase and be more precise.

L28: may be located

L34-L39: please say a few words regarding the intrinsic resistance of various Candida species other than C. albicans, as this is an important differentiating factor in clinical practice.

„Gram-positive” and „Gram-negative” is the correct way of writing, use it throughout the paper!

L47: please consider including the following reference:https://pubmed.ncbi.nlm.nih.gov/31052511/

L49: what do you mean by increased virulence?

The sentence at line 24 was rephrased to: “Within the human body, microorganisms mostly exist in complex communities including bacteria, fungi and viruses.”

The sentence at line 26 was rephrased to: “In various niches of the host, interactions between fungi and bacteria frequently occur during infections.”

On line 28, “can be located” was corrected to “may be located” and “Gram-positive” and “Gram-negative” were adjusted throughout the manuscript.

We have added the reference to the paper “The continuing threat of methicillin-resistant Staphylococcus aureus” in line 58.

We have added a small paragraph on the intrinsic resistance of Candida species at line 36-40.

Finally, “increased virulence” indicates the increased ability of the involved microorganisms to cause disease and to overcome host defenses. In this context “increased pathogenicity” is probably more correct and we have adjusted this throughout the manuscript.

Section 2.

In some parts of this section, the authors are merely recounting the results of other authors without putting them in context or sythesizing them/generating new ideas.

The correct way to write is „gentamicin” (cf. tobramycin, this depends on the originator)

We have put the results of the studies in context at the end of each paragraph of section 2 and in addition, we have corrected “gentamicin” throughout the manuscript.

Section 3.

In this section, please include additional discussion on certain areas in light of the following references: https://pubmed.ncbi.nlm.nih.gov/32392793/, https://www.ncbi.nlm.nih.gov/pmc/articles/PMC6429336/

Table 1. is a bit disorganized and hard to read, the authors should make an effort to improve clarity

3.2. please discuss the possibility of using the drug repurposing strategy to find novel QS-inhibitors: https://pubmed.ncbi.nlm.nih.gov/31861228/

3.8. the authors should expand on more relevant strategies. e.g, if biofilm-ihibition in concerned, bacteriophages were shown to be effective

Other: there was an unreasonable amount of emphasis on oral infections and the oral microflora. Would it be relevant to also explore other anatomical areas/infectious pathologies where polymicrobial infections are relevant?

We have added a few words on the application of photodynamic therapy in dentistry in section 3.4 (line 512-515), on the possibility of drug repurposing for the identification of new quorum quenching molecules in section 3.2 (line 439-442) and on the implications for the development of new drugs in the conclusion section (line 784-788) and we have added the proposed references.

In addition, we have added a small paragraph on the possibility to use bacteriophages and phage therapy in combination with antibiotics against fungal-bacterial biofilms in section 3.8 (line 746-756).

We have re-organized table 1 to have an easier to read overview.

Finally, the oral microflora is studied the most extensively in the context of fungal-bacterial interactions and is therefore emphasized in this manuscript. However, it is off course of interest to study polymicrobial infections in other pathologies as well. Therefore, we have added a few words on the lack of research in other anatomical areas in the conclusion section.